# Production and Characterization of Anti-Inflammatory *Monascus* Pigment Derivatives

**DOI:** 10.3390/foods9070858

**Published:** 2020-07-01

**Authors:** Deokyeong Choe, Soo Min Song, Chul Soo Shin, Tony V. Johnston, Hyung Jin Ahn, Daehwan Kim, Seockmo Ku

**Affiliations:** 1Fermentation Science Program, School of Agriculture, College of Basic and Applied Sciences, Middle Tennessee State University, Murfreesboro, TN 37132, USA; deokyeong.choe@mtsu.edu (D.C.); tony.johnston@mtsu.edu (T.V.J.); 2Department of Biochemistry and Cancer Biology, Meharry Medical College, Nashville, TN 37208, USA; 3Department of Biotechnology, College of Life Science and Biotechnology, Yonsei University, Seoul 03722, Korea; soomini16@nate.com (S.M.S.); csshin@yonsei.ac.kr (C.S.S.); 4Department of Food and Nutrition, Research Institute of Human Ecology, Seoul National University, Seoul 08826, Korea; ahnjin2000@snu.ac.kr; 5Department of Biology, Hood College, Frederick, MD 21701, USA

**Keywords:** amine derivatives, amino acid derivatives, anti-inflammatory effects, submerged fermentation, orange *Monascus* pigments

## Abstract

The prevention and treatment of chronic inflammation using food-derived compounds are desirable from the perspectives of marketing and safety. *Monascus* pigments, widely used as food additives, can be used as a chronic inflammation treatment. Orange *Monascus* pigments were produced by submerged fermentation in a 5 L bioreactor, and multiple orange *Monascus* pigment derivatives with anti-inflammatory activities were synthesized using aminophilic reaction. A total of 41 types of pigment derivatives were produced by incorporating amines and amino acids into the orange pigments. One derivative candidate that inhibited nitric oxide (NO) production in Raw 264.7 cells and exhibited low cell cytotoxicity was identified via in vitro assay. The 2-amino-4 picoline derivative inhibited NO production of 48.4%, and exhibited cell viability of 90.6%. Expression of inducible NO synthase, an important enzyme in the NO synthesis pathway, was suppressed by such a derivative in a dose-dependent manner. Therefore, this derivative has potential as a functional food colorant with anti-inflammatory effects.

## 1. Introduction

Inflammation is a protective response of the body to noxious stimuli, such as infection, tissue injury, and irritants [1,2,3]. It plays a role in eliminating the initial cause of cell injury, protecting cells from the spread of infection, initiating tissue repair, and restoring tissue function [4,5]. However, chronic inflammation, which is a prolonged abnormal inflammatory response [6], can cause many diseases including cancer, diabetes, cardiovascular disease, autoimmune disease, osteoarthritis, depression, and Alzheimer’s disease [7,8,9]. The World Health Organization (WHO) has reported that the greatest threat to public health is chronic inflammation and its associated diseases [10]. Moreover, 60% of the global mortality rate is attributed to chronic inflammatory diseases [11].

To overcome the problem of chronic inflammation, various types of anti-inflammatory drugs have been developed, and studies on newer anti-inflammatory drugs are ongoing. Anti-inflammatory drugs (e.g., glucocorticoids) are one of the effective approaches to treating chronic inflammation, but prolonged or high-dose glucocorticoid therapy has multiple side effects [12]. With these safety concerns, natural foods with anti-inflammatory properties have received attention as alternative strategies for the prevention and treatment of chronic inflammation. Alleviating inflammation and strengthening the immune system are the main benefits of anti-inflammatory foods such as probiotics, blueberry, tomato, avocado, salmon, olive oil, garlic, ginger, green tea, almond, spinach, oat, broccoli, and onion [13,14,15]. The increasing interest in anti-inflammatory foods has also boosted research on natural functional pigments with anti-inflammatory effects because such pigments can be used as food coloring agents, in addition. The best-known examples of natural functional pigments with anti-inflammatory properties are quercetin [16,17], curcumin [18,19], anthocyanin [20,21], and *Monascus* pigments [22,23].

*Monascus* pigments, which are microbial colorants, are secondary metabolites produced by the fermentation of edible *Monascus* species fungi [24,25]. *Monascus* pigments are a complex mixture of compounds with an azaphilone skeleton, which is a class of a typical fungal polyketide pigment [26,27]. *Monascus* pigments have been used as food additives in Asian countries for thousands of years [28]. They are traditionally classified as red, orange, and yellow pigments based on their maximum absorbance [29]. Among these three colors, orange pigments can be transformed into *Monascus* pigment derivatives by aminophilic reaction [30]. Specifically, pyranyl oxygen in orange pigments is replaced with a primary amine such as a protein, peptide, amino sugar, amino alcohol, or nucleic acid, and subsequent derivatives with red color are formed [31]. In recent years, various derivatives of orange pigments have been produced with amines and amino acids in our laboratory. These derivatives have exhibited a variety of biological properties, such as antimicrobial activity [32], hepatitis C virus replication inhibition [33], anti-atherosclerosis effects [34], diet-related lipase and α-glucosidase inhibitory activities [35], cholesteryl ester transfer protein inhibitory activity [36], obesity inhibitory activity [37], and melanogenesis inhibition [38]. However, to the best of our knowledge, there are few reports on the evaluation of *Monascus* pigment derivatives for anti-inflammatory effects.

In this study, orange *Monascus* pigments produced through submerged fermentation were converted to various amine and amino acid derivatives through bioprocessing. One *Monascus* pigment derivative that strongly inhibits nitric oxide (NO) production in cells and shows low cytotoxicity was selected. Its inhibitory effect was further analyzed for the expression of an enzyme involved in the inflammatory response.

## 2. Materials and Methods

### 2.1. Materials

Amines, amino acids, silica gel, fetal bovine serum (FBS), lipopolysaccharide (LPS), 3-(4,5-dimethylthiazol-2-yl)-2,5-diphenyltetrazolium bromide (MTT), protease inhibitor cocktail (P2714), and dimethyl sulfoxide (DMSO) were purchased from Sigma-Aldrich Co. (St. Louis, MO, USA). Ethyl acetate, n-hexane, ethanol, methanol, acetonitrile, potassium hydrogen phosphate, potassium dihydrogen phosphate, sodium nitrate, magnesium sulfate heptahydrate, iron(II) sulfate heptahydrate, sodium hydroxide, hydrochloric acid, acetic acid, potassium chloride, sodium chloride, sucrose, glucose, and agar powder were purchased from Duksan Co. (Seoul, Korea). Ammonium nitrate was purchased from Samchun Chemicals Co. (Seoul, Korea). Bacto-peptone, casamino acid, and yeast extract were purchased from BD Difco (Franklin Lakes, NJ, USA). Raw 264.7 cells (a murine macrophage cell line) were obtained from the Korea Cell Line Bank (Seoul, Korea). Dulbecco’s modified Eagle medium (DMEM) were purchased from Gibco (Grand Island, NY, USA). Antibodies for inducible nitric oxide synthase (iNOS) and α-tubulin were purchased from Santa Cruz Biotechnology, Inc. (Dallas, TX, USA). Reagents for Western blotting were purchased from ELPIS Biotechnology (Seoul, Korea). Griess reagent was purchased from Promega (Madison, WI, USA).

### 2.2. Cells and Media

*Monascus* sp. KCCM 10093 (Korea Culture Center for Microorganisms; Seoul, Korea) was used to produce orange *Monascus* pigments. The strain was preserved on a slant of Hiroi agar medium, which consisted of sucrose (100 g/L), casamino acid (5 g/L), yeast extract (3 g/L), NaNO_3_ (2 g/L), KH_2_PO_4_ (1 g/L), MgSO_4_·7H_2_O (0.5 g/L), KCl (0.5 g/L), FeSO_4_·7H_2_O (0.01 g/L), and agar powder (20 g/L) in distilled water. Mizutani medium for spore culture consisted of glucose (50 g/L), Bacto-peptone (20 g/L), KH_2_PO_4_ (8 g/L), CH_3_COOH (2 g/L), NaCl (1 g/L), and MgSO_4_·7H_2_O (0.5 g/L) in distilled water. The fermentation culture medium for orange pigment production consisted of glucose (50 g/L), NH_4_NO_3_ (3 g/L), KH_2_PO_4_ (1 g/L), MgSO_4_·7H_2_O (0.5 g/L), KCl (0.5 g/L), and FeSO_4_·7H_2_O (0.01 g/L) in distilled water. All media were adjusted to pH 6.6 prior to sterilization.

Raw 264.7 cells were used to evaluate the inhibitory activities of *Monascus* pigment derivatives against nitric oxide (NO) production. The cell growth medium consisted of DMEM (phenol red-free) supplemented with streptomycin (100 μg/mL), penicillin (100 U/mL), and 10% FBS.

### 2.3. Procedures for Monascus Cultivation

For the preparation of spore suspensions, *Monascus* sp. KCCM 10093 was grown on Hiroi agar slants for 7 days at 30 °C. After 10 mL of sterilized distilled water was added to each slant, spores were scraped off using a sterile spatula. For the spore culture, spore suspensions were inoculated into 500 mL Sakaguchi flasks containing 75 mL of sterilized Mizutani medium, followed by incubation for 48 h on a reciprocal shaking water bath at 30 °C and 120 rpm. For the seed culture, the spore culture broth was inoculated at 7% (*v/v*) into 500 mL Sakaguchi flasks containing 75 mL of sterilized Mizutani medium, followed by incubation for 24 h on a reciprocal shaking water bath at 30 °C and 120 rpm. For the fermentation culture, the seed culture broth was inoculated at 7% (*v/v*) into a 5 L bioreactor containing 3 L of a sterilized fermentation medium. Cultivations were performed for 120 h at 30 °C and 500 rpm with an aeration rate of 1.0 vvm.

### 2.4. Extraction and Identification of Orange Pigments

To extract orange pigments, 3 L of fermentation broth was filtered with Whatman filter paper (No. 4), and the filter cake was mixed with 2 L of ethyl acetate. The mixture was incubated for 48 h on a reciprocal shaking water bath at 30 °C and 150 rpm. After separating the ethyl acetate layer containing orange pigments, the solution was condensed to about 100 mL using a rotary evaporator (N-1110V-WD, Eyela Rikakikai Company, Tokyo, Japan) at a constant temperature of 70 °C. Approximately 200 g of silica gel powder was added to the solution for the adsorption of pigments. The pigment–silica gel complex was mixed with 1 L of n-hexane, followed by incubation at room temperature for 10 h. The hexane layer containing impurities was then removed. This n-hexane treatment was repeated five times. For the dissolution of the pigment, 1 L of ethyl acetate was added to the pigment–silica gel complex, and the ethyl acetate layer containing orange pigments was filtered with Whatman filter paper (No. 4) to remove the residual silica gel. Ethyl acetate was removed from the solution using a rotary evaporator (N-1110V-WD, Eyela Rikakikai Company, Tokyo, Japan) at a constant temperature of 70 °C, and orange pigments were obtained in solid form.

Identification of the orange pigments was performed in accordance with our previous methods [36,39]. The orange pigments were analyzed by HPLC and LC-MS. An HPLC (Acme 9000, Younglin Instrument, Seoul, Korea) with a GOLD C18 column (Hypersil, 250 × 4.6 mm and 5 μm) was operated with a run time of 60 min, flow rate of 1.25 mL/min, and elution ratio of distilled water/acetonitrile of 40:60 (*v/v*). An LC-MS (UPLC-LTQ-Orbitrap XL, Thermo Fisher Scientific, Waltham, MA, USA) with an Acquity UPLC BEH C18 column (50 × 2.1 mm and 1.7 μm) was operated with a run time of 18 min, flow rate of 0.3 mL/min, and elution ratio of 0.1% formic acid/acetonitrile of 40:60 (*v/v*). The mass spectrometer was operated in the ESI-positive mode, and the spray voltage was 5 kV. The capillary voltage, tube lens voltage, and capillary temperature were 35 V, 100 V, and 370 °C, respectively.

### 2.5. Determination of Cell Mass, Dissolved Oxygen Tension, pH, and Orange Pigment Production

To measure the cell mass, culture broths were filtered through Whatman filter paper (110Φ, No. 4) and washed with distilled water to remove culture debris. Cell concentrations were calculated after the filter papers were incubated in a drying oven (FO-600M, Jeio Tech., Daejeon, Korea) for 24 h at 80 °C. Dissolved oxygen tension (DOT) was measured using a DO probe (D100, Broadley-James Corp., Irvine, CA, USA). The pH was measured using a pH probe (F-615, Broadley-James Corp., Irvine, CA, USA). The concentration of orange pigments was measured by HPLC (Acme 9000, Younglin Instrument, Seoul, Korea).

### 2.6. Procedures for the Synthesis of Monascus Pigment Derivatives

The synthesis of *Monascus* pigment derivatives was performed in accordance with our previous methods [31,38]. Orange pigments, amines, and amino acids were individually dissolved in ethanol. Equal volumes of orange pigment solution and nitrogenous solution (amine or amino acid solution) were mixed together, and the mixture was agitated with a vortexer, followed by incubation at 60 °C for 1 h. In the case of amino acid derivatives, potassium phosphate buffer (3 M, pH 9.7) was added to the mixture to promote the synthesis reaction. After the completion of the reaction, the buffer was removed from the derivative solution through layer separation. Thus, 21 amine derivatives and 20 amino acid derivatives were synthesized (Table 1 and Figure 1a).

### 2.7. Measurement of NO Concentration

Raw 264.7 cells were maintained in DMEM (phenol red-free) supplemented with streptomycin (100 μg/mL), penicillin (100 U/mL), and 10% FBS, and cultivated at 37 °C in a humidified incubator with 5% CO_2_. To assess the NO levels, nitrite concentration was measured as an indicator of NO production. Raw 264.7 cells (2 × 10^5^/well) were plated and treated with LPS (10 μg/mL) in the presence or absence of *Monascus* pigment derivatives (20 μM). After incubation for 18 h, the suspended media were collected. The nitrite concentration in the culture medium was measured using a Griess reaction. A total of 100 μL of the sample supernatants was mixed with 100 μL of the Griess reagent (1% sulfanilamide in 5% phosphoric acid and 0.1% naphthylethylenediamine dihydrochloride in water). Absorbance of the mixture at 560 nm was determined using a microplate reader. After preparing a standard curve using NaNO_3_, the nitrite concentration was determined by comparing the value of absorbance.

### 2.8. MTT Assay for Cell Viability

The cell viability of *Monascus* pigment derivatives was measured using an MTT assay. Raw 264.7 cells were seeded in 96-well plates (2 × 10^5^/well) and incubated for 24 h in a humidified 5% CO_2_ incubator at 37 °C. The derivatives (20 μM) were added into the wells, and the plates were incubated for 24 h. For the control, the derivative was not added. After supernatants were removed, 100 μL of MTT solution (1 mg/mL) was added to the cells. The plates were incubated for 3 h, followed by the removal of the MTT solution. Then, 100 μL of DMSO was added to the wells to dissolve formazan. The absorbance was determined using a microplate reader at a wavelength of 560 nm.

### 2.9. Western Blot Analysis

Raw 264.7 cells (2 × 10^5^/well) were lysed with a radio-immunoprecipitation assay buffer containing protease and phosphatase inhibitors on ice for 10 min. The lysates were centrifuged at 13,800× *g* for 20 min. The protein concentration of the cell lysates was determined using Bradford reagent. The samples were subjected to sodium dodecyl sulfate-polyacrylamide gel electrophoresis (SDS-PAGE) for 2 h and transferred to the nitrocellulose membranes for 2 h. The membranes were blocked with 5% non-fat milk in Tris-buffered saline with 1% Tween 20 (TBST). After washing with TBST, the membranes were incubated with primary antibodies specific for iNOS and α-tubulin at 4 °C overnight, followed by washing with TBST. Blots were probed with goat anti-rabbit immunoglobulin G conjugated to peroxidase for 1 h. Immunoreactive bands were detected by an electrochemiluminescence detection system (Syngene, Cambridge, UK).

## 3. Results and Discussion

### 3.1. Production of Orange Monascus Pigments through Submerged Fermentation

To synthesize *Monascus* pigment derivatives, their basic building blocks, orange *Monascus* pigments, are required. As red and yellow *Monascus* pigments are not involved in the aminophilic reaction for the synthesis of *Monascus* pigment derivatives [30], maximizing the yield of orange pigments by suppression of the production of red and yellow pigments during the whole process is an adequate approach. To achieve this goal, inorganic ammonic compounds (e.g., ammonium nitrate, NH_4_NO_3_) as a nitrogen source for *Monascus* fermentation were used in our previous studies [31,39,40]. However, there are few reports on the effects of ammonium nitrate on the production of orange pigments and on the submerged culture of *Monascus* sp. KCCM 10093. Investigating these effects was perceived to provide useful information to improve the productivity of orange pigments, so we evaluated cell mass, DOT, pH, and orange pigment yield by monitoring a 5 L bioreactor at 24 h intervals for 5 days (Figure 2).

Cell mass rapidly increased up to 8.0 g/L within 48 h and continuously increased for 120 h, showing a final cell concentration of 9.9 g/L. With increasing cell mass, DOT consistently decreased from 95% to 40%, indicating that oxygen was almost uniformly consumed during the whole fermentation process. Aerobic microorganisms require an abundance of oxygen for cell growth during the log phase, whereas continuous oxygen consumption was observed during the whole fermentation process for the production of orange pigments. This result implies that oxygen is necessary not only for cell growth but also for orange pigment production. Indeed, the production of orange pigments was reported to be reduced by the limitation of dissolved oxygen [41], and an improvement in oxygen supply led to an increase in *Monascus* pigments [42,43,44]. In addition, oxygen is a substrate of a flavin adenine dinucleotide-dependent monooxygenase, which is involved in a biosynthetic pathway for orange pigments [29,45,46]. Therefore, we suggest that oxygen is one of main factors in the production of orange pigments through submerged fermentation.

Unlike the observation of continuous oxygen consumption, the growth rates (red and blue lines in Figure 2), which are the change in the cell mass per unit time, were considerably different before and after 48 h. This difference was probably due to the change in pH. The pH values were 7.2, 4.3, 3.2, 2.9, 2.7, and 2.6 at 0, 24, 48, 72, 96, and 120 h, respectively. In other words, the initial pH of 7.2 rapidly decreased by 3.2 units within 48 h and then further decreased by 2.6 more units. Such pH reduction was caused by ammonium nitrate [47]. When ammonium (NH_4_^+^) salts are added as a nitrogen source, *Monascus* sp. assimilates ammonia (NH_3_) and releases protons (H^+^), leading to acidification of the culture broth [48]. Low pH can play a role in the inhibition of cell growth, as the preferred pH for *Monascus* cell growth is 5.5–6.5 [44,49]. In contrast to growth rates, the synthesis of orange pigments was promoted by low pH. For the 0–48 h period showing a high growth rate and high pH, the orange pigment yield was 0.6 g/L. For the 48–120 h period showing a low growth rate and low pH, the production of orange pigments rapidly increased and the final yield reached 4.1 g/L at 120 h (Figure 2). This result is in agreement with previous reports in which large amounts of orange pigments were obtained under acidic fermentation conditions [50,51], red pigments were mainly produced at pH 6.5, and orange pigments were observed in pH 2.5 media [44,49]. The increase in orange pigment yield under low-pH conditions can be caused by inhibition of the transformation of orange to red pigments [28,52]. As red pigments are known to be synthesized by amination reaction from orange pigments [29,52], inhibition of this transformation can result in a substantial increase in orange pigments during the fermentation process. Furthermore, at low pH, a significant increase in the expression of key enzyme-coding genes related to the biosynthesis of orange pigments, such as *MrpigA*, *MrpigB*, *MrpigF*, *MrpigJ*, and *MrpigK*, has been reported [50]. Their increased expression contributed to the enhancement in orange pigment yields. Consequently, the use of ammonium nitrate as a nitrogen source for *Monascus* fermentation reduced the pH, leading to enhanced production of orange pigments.

### 3.2. Preparation Process for Orange Monascus Pigments and Synthesis of Derivatives

Orange *Monascus* pigments produced through submerged fermentation were extracted, condensed, separated, and purified in the presence of ethyl acetate, n-hexane, or silica gel (Figure 3). When the fermentation was completed, a mixture of culture broth, mycelia, and orange pigments was obtained from the 5 L bioreactor. As orange pigments are insoluble in water due to their hydrophobicity [30,53], they were separated from the culture broth by filtration, which resulted in the creation of filter cakes laden with orange pigments. To extract the orange pigments from the filter cakes, a solvent with a polarity suitable for dissolving the orange pigments was required. Of the organic solvents permitted for use in the food industry (methanol, ethanol, ethyl acetate, acetone, and n-hexane) [54], ethyl acetate has been used for the extraction of orange pigments in previous studies [55,56]. Orange pigments are highly soluble in ethyl acetate, an amphiphilic volatile organic solvent. It can effectively extract the targeted orange pigments without chemical changes and/or damage to the pigments. Moreover, ethyl acetate is generally recognized as safe (GRAS) for use in beverages and foods as a flavoring agent and adjuvant [57]. For these reasons, ethyl acetate was selected as the extraction solvent for orange pigments in this study.

After extraction of orange pigments from the filter cakes, the color of the ethyl acetate changed from transparent to dark red, indicating that the orange pigments were already dissolved in the solvent. Undissolved culture debris was discarded to improve the purity of the pigments, and the resulting solution was condensed by evaporation. For further purification, the condensed solution was mixed with silica gel, and a pigment–silica gel complex with dark red color was obtained. To remove hydrophobic impurities from the orange pigments, the pigment–silica gel complex was extracted by n-hexane, a lipophilic volatile organic solvent that could dissolve undesirable hydrophobic substances but poorly extract orange pigments [28]. After completion of the n-hexane extraction, pure ethyl acetate was again used to obtain an orange pigment solution with enhanced purity from the pigment–silica gel complex. The silica gel remaining in the orange pigment solution was eliminated by filtration, and the ethyl acetate was evaporated during the condensing process. The purified orange pigments were then obtained in solid form.

The purified orange pigments were used as the basic building blocks for the synthesis of many types of *Monascus* pigment derivatives. As shown in Figure 1a, each derivative was synthesized by aminophilic reaction, in which the pyranyl oxygen in the orange pigments was replaced with the nitrogen of the amino group of primary amines or amino acids [28,30]. This synthetic method can induce the production of diverse *Monascus* pigment derivatives with novel molecular structures and increase the possibility of each derivative having different functionalities, such as antimicrobial, antiviral, anticholesterol, antiobesity, and antimelanogenesis effects [31,58]. In addition to these functional effects, 21 amines and 20 amino acids were added to the orange pigments to obtain anti-inflammatory effects, leading to the synthesis of 41 derivatives (Table 1).

### 3.3. NO Production and Cytotoxicity of Monascus Pigment Derivatives in Raw 264.7 Cells

Measurement of nitric oxide (NO) levels is a common method for assessing the anti-inflammatory capabilities of bioactive compounds. NO is secreted by macrophages (e.g., Raw 264.7 cells) when exposed to inflammatory stimuli such as lipopolysaccharide (LPS) embedded in the outer membrane of Gram-negative bacteria [17,59]. Suppression of LPS-induced NO production in Raw 264.7 cells is considered evidence of potential anti-inflammatory capability. In other words, the lower the NO level in treated cells, the higher the anti-inflammatory activity of the compound. To evaluate the anti-inflammatory effects of *Monascus* pigment derivatives synthesized in this research, the NO levels in Raw 264.7 cells after exposure were determined and compared.

The NO production of cells treated with the 21 amine derivatives was 48.8–90.2% when the NO level of LPS-stimulated cells (control cells) with orange *Monascus* pigments was set to 100% (Table 2). In particular, the NO levels of cells treated with D2, D5, D8, D14, D15, D16, D19, and D20 derivatives showed statistically significant differences compared to the control cells. However, the D5, D8, D14, D15, D16, D19, and D20 derivatives exhibited statistically low cell viability (Table 2). This indicates that their low NO production was caused by the cytotoxicity. Similar tests were performed on the 20 amino acid derivatives, which resulted in an NO production of 84.3–147.2%. The NO levels of cells treated with amino acid derivatives were not statistically different from the control cells. This does not indicate that amino acid derivatives have no anti-inflammatory effect. Rather, this shows that their NO inhibitory effects are similar to those of orange *Monascus* pigments, which are known to have anti-inflammatory effects [60,61]. The NO inhibitory effects of these amine and amino acid derivatives are due to the azaphilone skeleton. Previous studies reported that *Monascus* pigments with the azaphilone structure showed anti-inflammatory activities [22,60]. Specifically, azaphilone *Monascus* pigments have been shown to exhibit much stronger anti-inflammatory effects than non-azaphilone *Monascus* pigments [62]. Based on our results and previously published data, we conclude that amine and amino acid derivatives with the azaphilone skeleton have anti-inflammatory properties.

Despite having the same azaphilone structure, the amine derivatives showed a higher efficacy in suppressing NO production than amino acid derivatives (Table 2). This difference was likely caused by the amines and amino acids used for the synthesis of the *Monascus* pigment derivatives. Unlike amino acids, most amine compounds contain structures such as pyridine, triazole, and phenyl rings, which may contribute to increased anti-inflammatory effects. Flavonoids, including anthocyanins, catechins, flavones, isoflavones, and flavonols, consist of multiple phenyl groups and are widely recognized as natural anti-inflammatory agents [63,64]. We suggest that the combination of the azaphilone skeleton-containing orange *Monascus* pigments and the ring structure-containing amines into the amine derivatives leads to higher anti-inflammatory effects compared to the amino acid derivatives.

To determine their potential application as anti-inflammatory food colorants, the toxicity of the amine and amino acid derivatives was tested for safety. To assess cytotoxicity, cell viability was determined using the MTT assay with Raw 264.7 cells. Most of the amine derivatives showed low cell viabilities at 20 µM concentration, which was used to inhibit NO production (Table 2). Compared to the control cells treated with orange *Monascus* pigments, a decrease in cell viability of less than 80% was observed in 19 of the 21 amine derivatives, indicating that most of the amine derivatives were toxic to Raw 264.7 cells. However, two amine derivatives, D2 and D12, and the 20 amino acid derivatives showed high cell viabilities (>80%), indicating their negligible cytotoxicity (Table 2). It is worthwhile to mention that these cell viability data are opposite to our previous results [39] for cytotoxicity, where cell viabilities were 87–92% when 4-phenylbutylamine derivatives were treated to 3T3-L1. It is possible that a different cell was used for tests with a different target (3T3-L1 cell for lipid adipocyte vs. Raw 264.7 cell for nitric oxide production) that showed a different cell viability.

As previously described, the amino acid derivatives were synthesized by the addition of amino acids, which are naturally produced organic compounds within the cells of living organisms. The 19 amino acids, except theanine (D41), are known to be key molecules for building proteins (Table 1). Owing to their low toxicity, origin, and role, amino acids are considered as GRAS [65]. The high safety of amino acids was probably imparted to the amino acid *Monascus* pigment derivatives, contributing to their high cell viabilities. In contrast to amino acids, the amines used for the synthesis of the amine derivatives are mainly synthesized through artificial chemical reactions and are not natural compounds found in living organisms. For this reason, amines generally have high toxicity. This toxicity was apparently imparted to the amine derivatives, resulting in low cell viability. However, some amine derivatives are not toxic at low administration levels. As reported in our previous study, two amine derivatives of the *Monascus* pigment, (R)-(+)-1-(1-naphtyl)-ethylamine and 4-phenylbutylamine, showed high cell viabilities (>80%) in 3T3-L1 cells up to 25 µM concentration [39]. Similarly, in this study, one amine derivative, D2, also exhibited high cell viability (89.7%) in Raw 264.7 cells. Considering these results, we conclude that the D2 derivative has the potential for use due to its high anti-inflammatory activity and low cytotoxicity.

### 3.4. Inhibition of iNOS Expression in the Concentration of Amine Derivatives

Based on NO inhibitory activity and cell viability screening, the D2 derivative was selected. The D2 derivative showed statistically significant NO inhibitory activity (48.4%) and its cytotoxicity (89.7% cell viability) was not statistically different from the control cells (Table 2). To further evaluate its anti-inflammatory activity, the expression levels of iNOS were investigated by Western blot analysis. iNOS catalyzes NO formation from molecular oxygen (O_2_) and arginine [66], and it plays a role in regulating the release of NO [67]. Thus, suppression of LPS-induced iNOS expression is considered a potential anti-inflammatory effect.

As shown in Figure 4, iNOS was predominantly expressed in the LPS-stimulated Raw 264.7 cells without the D2 derivative, whereas expression was gradually reduced with increasing concentration of the D2 derivative. This indicates that the D2 derivative suppresses LPS-induced iNOS expression in a dose-dependent manner. In the case of the D2 derivative, the inhibition of iNOS expression began at a concentration of 2.5 μM and was almost completely suppressed at a concentration of 20 μM. This result is in agreement with previous results that showed that the D2 derivative has high cell viability and significantly inhibited NO production (Table 2). Considering an association between NO production and iNOS, we suggest that the inhibitory effect of the D2 derivative against NO production resulted from their iNOS suppression.

Similar to the D2 derivative, there are several reports of iNOS inhibition by other *Monascus* pigments. Monaphilone A and ankaflavin (yellow pigments), which are extracted from red mold rice, decreased iNOS expression at a concentration of 10 μM [68]. Furthermore, monaphilones A and B (yellow pigments) and monaphilols A–D (orange pigments), which were isolated from red mold rice and red mold dioscorea, respectively, suppressed iNOS expression at concentrations of 30 and 5 μM, respectively [61]. These *Monascus* pigments, like the D2 derivative, have azaphilone skeletons. Their structural commonality perhaps contributed to the inhibition of iNOS expression. On the other hand, despite having similar inhibitory effects, their colors are different (e.g., yellow and orange pigments). This variety of colors has practical implications for their application range as food colorants. Notably, because the D2 derivative is a *Monascus* pigment with red color, it has the potential to serve as a food colorant that could replace existing red pigments. To the best of our knowledge, no red pigments previously reported among *Monascus* pigments have anti-inflammatory properties. Therefore, the D2 derivative, which is characterized by a red color, high cell viability, and anti-inflammatory activity, could be used as a functional food colorant for the prevention and treatment of chronic inflammation.

## 4. Conclusions

Orange *Monascus* pigments were produced through submerged fermentation in a 5 L bioreactor. During the fermentation process, the use of ammonium nitrate as a nitrogen source resulted in low pH values, which increased the total yield of orange pigments by inhibiting the production of undesired red and yellow pigments. After separation and purification processes using ethyl acetate, n-hexane, and silica gel, 41 different *Monascus* pigment derivatives were synthesized from purified orange pigments by aminophilic reaction with 21 amines and 20 amino acids. Among the derivatives, one amine derivative, D2, significantly inhibited NO production and had little effect on cell viability. In addition, the D2 derivative suppressed iNOS expression in a concentration-dependent manner, indicating that it has anti-inflammatory properties. These results show that anti-inflammatory properties can be improved by structural modification (derivatization) of orange *Monascus* pigments with amines or amino acids.

## Figures and Tables

**Figure 1 foods-09-00858-f001:**
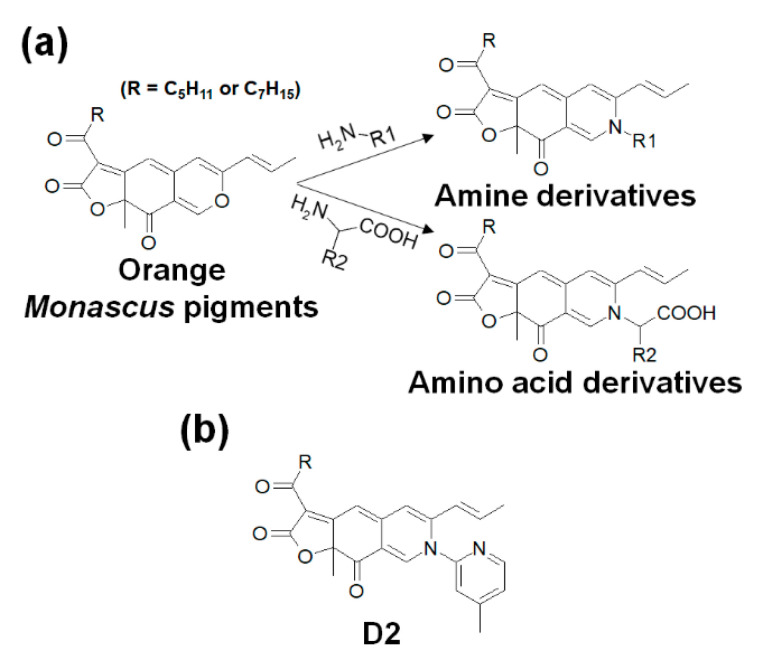
Synthesis of amine and amino acid derivatives from orange *Monascus* pigments (**a**) and the molecular structure of the 2-amino-4-picoline (D2) derivative of *Monascus* pigment (**b**).

**Figure 2 foods-09-00858-f002:**
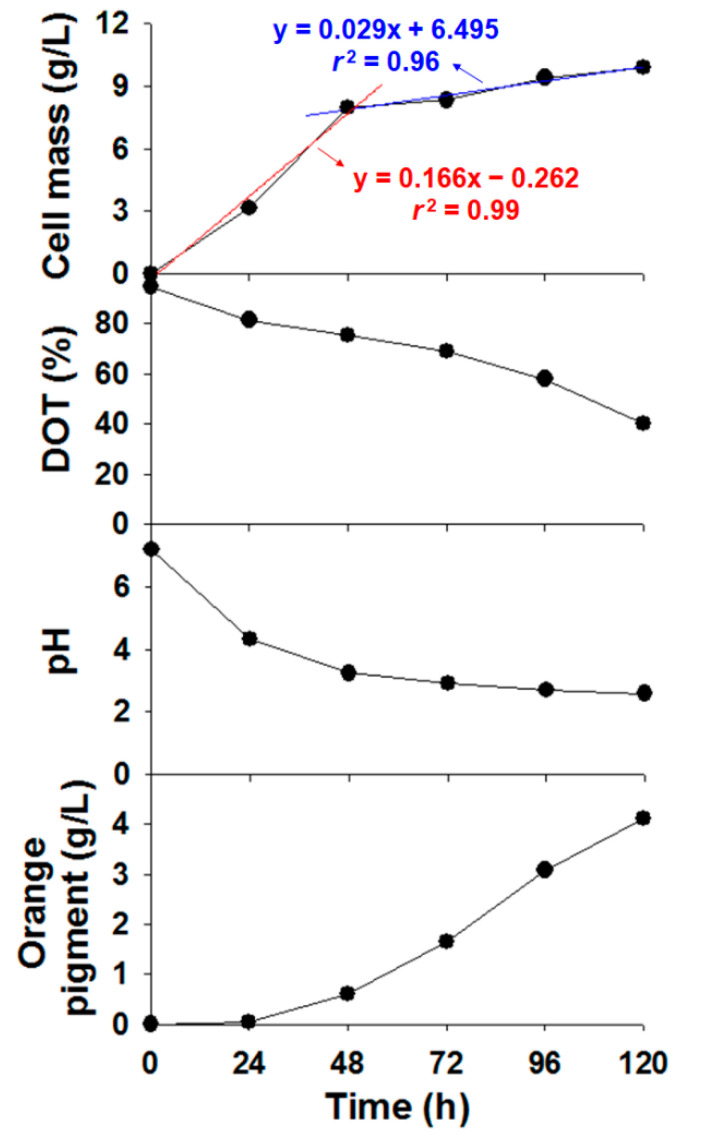
Production profiles of orange *Monascus* pigments in a 5 L bioreactor. DOT = dissolved oxygen tension.

**Figure 3 foods-09-00858-f003:**
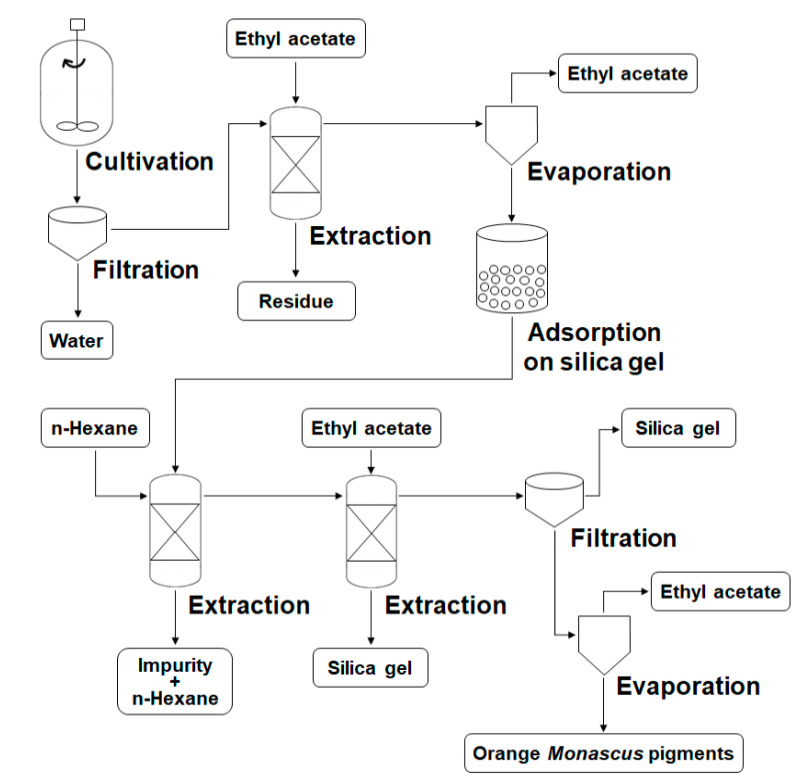
Process flow diagram for the separation and purification of orange *Monascus* pigments.

**Figure 4 foods-09-00858-f004:**
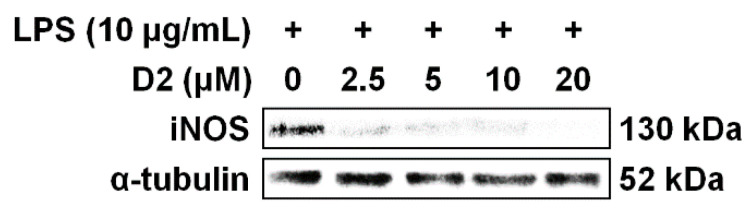
Concentration-dependent inhibitory effects of the 2-amino-4-picoline (D2) derivative of *Monascus* pigment on expression of inducible nitric oxide synthase (iNOS) in Raw 264.7 cells stimulated with lipopolysaccharide (LPS).

**Table 1 foods-09-00858-t001:** Amines and amino acids used for synthesis of *Monascus* pigment derivatives.

Derivative Compound	Amines	Derivative Compound	Amino Acids
D1	2-amino-6-methylpyridine	D22	Serine
D2	2-amino-4-picoline	D23	Threonine
D3	(S)-(+)-2-amino-1-propanol	D24	Cysteine
D4	5,6,7,8-tetrahydro-2-naphthylamine	D25	Methionine
D5	2-amino-5-bromopyridine	D26	Asparagine
D6	1-amino-4-methylpiperazine	D27	Glutamine
D7	2-amino-4-chlorophenol	D28	Aspartic acid
D8	2-amino-5-methylthiazole	D29	Glutamic acid
D9	2-amino-5-chlorophenol	D30	Lysine
D10	2-amino-4-fluorophenol	D31	Arginine
D11	2-amino-5-iodopyridine	D32	Histidine
D12	4-amino-4H-1,2,4-triazole	D33	Phenylalanine
D13	Methyl-d_3_-amine hydrochloride	D34	Tyrosine
D14	(R)-(-)-1-amino-2-propanol	D35	Tryptophan
D15	(S)-(+)-1-amino-2-propanol	D36	Glycine
D16	5,6,7,8-tetrahydro-4H-cyclohepta[d][1,3]thiazol-2-amine	D37	Alanine
D17	Benzylamine	D38	Valine
D18	4-methylphenethylamine	D39	Leucine
D19	4-aminotetrahydropyran	D40	Isoleucine
D20	(R)-(-)-2-amino-1-butanol	D41	Theanine
D21	4-phenylbutylamine		

**Table 2 foods-09-00858-t002:** Nitric oxide (NO) production and cell viability of amine and amino acid derivatives in Raw 264.7 cells.

Amine Derivative	NO Production (%)	Cell Viability (%)	Amino Acid Derivative	NO Production (%)	Cell Viability (%)
Con	100.0 ± 7.1	100.0 ± 3.8	Con	100.0 ± 4.9	100.0 ± 4.0
D1	64.6 ± 3.4	53.9 ± 1.0 *	D22	143.0 ± 4.7 *	94.0 ± 5.5
D2	51.6 ± 0.9 *	89.7 ± 8.4	D23	106.9 ± 2.3	84.9 ± 6.4
D3	56.2 ± 0.5	31.3 ± 6.7 *	D24	147.2 ± 6.1 *	86.3 ± 5.6
D4	61.2 ± 2.0	12.9 ± 1.6 *	D25	100.6 ± 3.6	93.8 ± 2.6
D5	56.2 ± 3.1 *	58.4 ± 5.1 *	D26	135.2 ± 6.6 *	91.2 ± 5.0
D6	65.1 ± 2.0	13.5 ± 4.7 *	D27	106.6 ± 18.1	103.2 ± 5.3
D7	70.8 ± 1.7	3.5 ± 1.2 *	D28	126.6 ± 5.2	96.4 ± 15.8
D8	48.8 ± 0.5 *	70.9 ± 3.0 *	D29	103.1 ± 5.0	104.0 ± 6.3
D9	66.2 ± 2.9	2.4 ± 1.0 *	D30	105.5 ± 2.2	112.0 ± 4.5
D10	90.2 ± 2.5	2.5 ± 1.5 *	D31	95.8 ± 3.0	104.5 ± 0.8
D11	63.6 ± 11.4	73.3 ± 1.1 *	D32	126.4 ± 4.2 *	90.6 ± 8.8
D12	73.3 ± 24.3	86.7 ± 5.4	D33	97.3 ± 6.0	103.5 ± 4.9
D13	73.5 ± 15.4	60.4 ± 8.1	D34	141.7 ± 17.6	97.8 ± 2.1
D14	55.6 ± 2.4 *	23.4 ± 4.9 *	D35	103.0 ± 11.4	109.0 ± 4.5
D15	55.9 ± 3.3 *	63.7 ± 2.2 *	D36	137.3 ± 16.3	102.6 ± 1.0
D16	56.8 ± 2.9 *	16.3 ± 2.9 *	D37	100.9 ± 2.3	104.7 ± 7.0
D17	60.3 ± 11.3	44.5 ± 7.3 *	D38	118.3 ± 4.8	101.3 ± 3.4
D18	74.8 ± 5.7	12.0 ± 6.7 *	D39	100.3 ± 7.6	108.0 ± 2.9
D19	56.2 ± 4.0 *	9.6 ± 1.0 *	D40	136.7 ± 13.7	93.7 ± 4.3
D20	58.2 ± 2.6 *	6.3 ± 3.4 *	D41	84.3 ± 4.4	98.4 ± 3.3
D21	66.1 ± 3.1	7.3 ± 3.2 *			

One-way ANOVA followed by Tukey’s HSD and Games–Howell post-hoc tests were performed. Values are the mean ± SEM (n = 3). (*) *p* < 0.05 indicates significant differences compared to the control group. In the case of NO production, cells were treated with lipopolysaccharide (LPS) (10 μg/mL) in the presence or absence of *Monascus* pigment derivatives (20 μM). Con = LPS-stimulated control cells with orange *Monascus* pigments. In the case of cell viability, cells were treated with or without *Monascus* pigment derivatives (20 μM). Con = control cells treated with orange *Monascus* pigments.

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
