# Peer review of "Production and Characterization of Anti-Inflammatory Monascus Pigment Derivatives"

_foods, 2020, doi:10.3390/foods9070858_

Round 1

Reviewer 1 Report

attached

Author Response

Major comments:

  1. Line 26-28 “The derivatives of 2-amino-4 picoline and 4-amino27 4H-1,2,4-triazole exhibited NO inhibitory activities of 74.2% and 63.3%, and cell viabilities of 90.6% 28 and 87.6%, respectively.” It looks like that the derivatives inhibit rather the growth of the macrophages than NO production. In order to conclude the anti-inflammatory potency, the compounds should reduce NO production together with no cytotoxicity. However, such comparison is in this current version of manuscript almost impossible, I suggest the preparation of the table comparing both activities.

Reply: Thanks to the reviewer for the constructive comments. Following the reviewer’s suggestion, we have added NO inhibitory activity and cell viability in Table 2 (line 398).

The Table 2 is more informative, I prefer to include SEM and results of ANOVA into this table and remove figures 4 and 5 (at this moment the figures duplicate the results of the Table 2). Both chapters (NO production and viability) should be merged as there could be made no conclusion based on the results of just only one assay. After that, the conlusions:

Line 294 “The NO production of cells treated with the 21 amine derivatives was 24.4–45.2%...”

Line 296 “The amine derivatives significantly decreased NO production by 54.8–75.6%...” (Btw. This is the duplication of previous sentence)

should be changed as the lower production of NO was caused by the cytotoxicity (it means lower amount of tested cells).

This is not the case of amino derivatives, which were non toxic; however the duplication:

Line 297 “… amino acid derivatives, which resulted in NO production of 49.9–87.2%...”

Line 298 “…the amino acid derivatives reduced the NO production by 22.8–50.1%...”

Should be removed as well.

Reply: We appreciate the reviewer’s thoughtful comments. As requested by the reviewer, figures 4 and 5 were removed and both chapters were merged. Duplicate sentences were also removed. Furthermore, we have addressed the low NO production of amines in lines 296–299 as follows: “In particular, the NO levels of cells treated with D2, D5, D8, D14, D15, D16, D19, and D20 derivatives showed statistically significant differences compared to the control cells. However, the D5, D8, D14, D15, D16, D19, and D20 derivatives exhibited statistically low cell viability (Table 2). This indicates that their low NO production was caused by the cytotoxicity.”

  1. Why anti-inflammatory, non-toxic amino derivatives were excluded from the iNOS experiment? At least D41 as the most active amino derivative?

Reply: Thanks to the reviewer for pointing this out. Since the D41 derivative has lower NO inhibitory activity than other derivatives, it has been ruled out for further iNOS tests. This has been addressed in lines 364–365.

I suggest the addition of some rule into the manuscript for derivatives choosing. In my opinion, derivative D12 inhibits NO production about 63.3% and viability about 12.4%. Derivative 41 inhibits NO production about 50.1% and viability about 2.6%. It looks to me that both derivatives have the same potential, haven´t they? Similarly, derivatives D25, D31, D33, D37, D39 inhibit almost the half of NO production and were not chosen for further investigation neither that the inhibition of NO production was considered as statistically significant. It should be explained in the manuscript what was the limit or rule.

Reply: The reviewer is correct that our statement did not reflect clearly the limit of further tests. Per the reviewer’s suggestion, we have reworded the relevant rule for choosing derivatives in the manuscript (lines 357–359). Based on statistical significance, the D12 and D41 derivatives were ruled out and the D2 derivative was selected (Table 2).

  1. Figure 2 lacks error bars.

Reply: This is a similar question to Q5 from reviewer #1. Our responses to the following questions address this question differently.

Unfortunately, I don´t understand the answer. Maybe that it could be due to the fact that I don´t see the reviewer #1 questions/answers?

Reply: The reviewer is correct that data reproducibility and statistical analysis are not provided in the manuscript. Since this is a follow-up study based on our previous research to produce orange pigment, its various derivatives via microbial fermentation, and their biological activities (ref #31,32,34–37,39,40 and 58), data on cell mass, DOT, pH and orange pigment yield shown in Figure 2 are similar to that of our previous works. In addition, other publications show a similar tendency for the pH to decrease and orange pigment yield to increase (ref #47,48). Because of the extensive documentation of these phenomena in other published data, we have not specifically addressed these issues in this paper.

  1. As the other control, original pigment should be used in order to evaluate the effect of modification on biological activity of original compound.

Reply: Thanks for the very good observation. As suggested by the reviewer, we have added the NO production data to Figure 4.

Perfect, however, this should be realized for the viability as well. I suggest the addition of some conclusion/discussion part about the effect of specific modification on biological structure comparing the original structure and its derivatives.

Reply: This is a very good comment and suggestion. We followed the reviewer’s suggestion and changed Table 2 accordingly and added relevant texts in lines 402–404: “These results show that anti-inflammatory properties can be improved by structural modification (derivatization) of orange Monascus pigments with amines or amino acids.”

  1. Western blot could be used as a semi-quantitative method; however, it has to follow the standard procedure (like to determine the linear dynamic range of both control and target protein loading; both target and control protein should be quantified in each sample – D2 sample lacks tubulin, etc.). This validation should be part of the methodology.

Reply: The reviewer is correct in that both target and control protein should be quantified in each sample. Our raw data included the α-tubulin as below. However, because similar α-tubulin results were obtained, we intentionally omitted one tubulin result to clearly show the concentration-dependent suppression of iNOS by the D2 and D12 derivatives. Thus, we want to keep Figure 6 as is to make comparisons between samples easier. 

That is great! I suggest to replace Figure 6 for this more valuable figure. It is a pity not to show that the presented results are obtained by an adequate procedure.

Reply: Thank you for your comment and suggestion. We removed the result of D12 per the recommendation of reviewer #1 and added the D2 result only. It is also important to note that we realized the α-tubulin results for D2 and D12 were switched in the previous manuscript; we corrected this as below. We believe this revision more clearly presents our experimental approach and results.

Figure 4. Concentration-dependent inhibitory effects of D2 derivative on expression of iNOS in Raw 264.7 cells stimulated with LPS.

Reviewer 2 Report

The authors have addressed all the concerns raised by the reviewer. Thank you for clarifying all the issues. 

The following comments address some minor issues and are offered for improving the paper. 

The authors indicate in the review response that the previous studies have shown the differences in biological activity by orange pigment derivatives are caused by differences in the structure of functional groups such as amines and amino acids

Since the focus the research is derivatization of orange pigments, Ora (the new control added) is a more suitable and appropriate control when comparing and assessing anti-inflammatory properties and cell viability of derivatized pigments. 

Cell viability data still lacks the control (Ora-LPS stimulated cells with orange pigments without derivatization. Assuming that Ora exhibits the similar cell viability as Control (without pigments), the conclusions should be valid. 

Author Response

The authors have addressed all the concerns raised by the reviewer. Thank you for clarifying all the issues. 

The following comments address some minor issues and are offered for improving the paper. 

The authors indicate in the review response that the previous studies have shown the differences in biological activity by orange pigment derivatives are caused by differences in the structure of functional groups such as amines and amino acids

Since the focus the research is derivatization of orange pigments, Ora (the new control added) is a more suitable and appropriate control when comparing and assessing anti-inflammatory properties and cell viability of derivatized pigments. 

Reply: We thank the reviewer for the careful review and thoughtful suggestions. In agreement with the recommendation of the reviewer, we have changed Ora to a new control and added it in Table 2.

Cell viability data still lacks the control (Ora-LPS stimulated cells with orange pigments without derivatization. Assuming that Ora exhibits the similar cell viability as Control (without pigments), the conclusions should be valid. 

Reply: As you predicted, orange pigments exhibited similar cell viability to that of control cells without pigments. This is in agreement with previous reports that orange pigments did not show cytotoxicity against Raw 264.7 cells (ref #60). As suggested by the reviewer, we have added the cell viability data to Table 2 and changed Ora to a new control.

Round 2

Reviewer 1 Report

I have no comments.

This manuscript is a resubmission of an earlier submission. The following is a list of the peer review reports and author responses from that submission.

Round 1

Reviewer 1 Report

  1. Please add the pH of the Manascus fermentation run,
  2. The authors need to add more details on how evaporation is conducted (apparatus used, temperature, concentration factor based on initial and final volume, etc).
  3. How much silica gel powders were used to extract the pigments?
  4. Refer to Table 1 in Section 2.6 in materials and methods as table 1 gives the list of amines and amino acids used to produce the derivatives.
  5. How many fermentation runs were made especially for Figure 2? The data reproducibility and statistical analysis need to be included.
  6. The authors need to explain how ammonium nitrate, used as a nitrogen source, lowers the pH during the Monascus fermentation.
  7. What was the extraction and purification yield of orange pigments? What was the purity of the orange pigments purified?
  8. Hexane is used to remove undesirable hydrophobic substances from silica gel-pigments complex. Considering hexane is a strong apolar solvent, how the authors made sure the hexane did not chemically change or damage the pigments during this process?
  9. Replace “condensing” in Figure3 with “Evaporation”.
  10. Cell toxicity data from Reference 39 showed high cell viability for 4-phenylbutylamine derivatives when using 3T3-L1 cells, while the same derivatives showed a very low cell viability using Raw 264.7 cells in Figure 5 (D21). If the cell toxicity depends on the cells used, then how can the cell viability test data be applied in actual applications?

The following describes the main issues with the paper.

  1. The authors used ammonium nitrate as a nitrogen source to suppress the formation of red and yellow pigments while improving the production of orange pigments by lowering the pH of fermentation. The authors conclude that ammonium nitrate lowers the pH of fermentation and enhances the formation of orange pigments. To support this conclusion and verify the effect of ammonium nitrate and pH on orange pigments production, the authors need to include the profiles of red and yellow pigments in Figure 2 for comparison as well as the profiles of red, orange and yellow pigments in a control run without ammonium nitrate. Without these data the conclusion is not supported by the data.
  2. Monascus azaphilone pigments are a complex mixture of compounds with a common azaphilone skeleton. They are classified as red, orange, and yellow pigments based on their absorbance profiles at specific wavelengths. The authors conclude that the anti-inflammatory effects are due to the azaphilone skeleton of the orange pigment derivatives. Given that Monascus pigments (red, yellow and orange) have a common azaphilone skeleton, what do amine and amino acids functional groups have to do with the observed anti-inflammatory effects? Would the pigments exhibit a similar anti-inflammatory effect without the derivatization with amine or amino acids? To support the conclusion with valid arguments, the authors need to give the NO production data of orange pigments without derivatization in Figure 4 for comparison. Also, red and yellow pigments have azaphilone skeleton and thus may possess anti-inflammatory effects. Then why are red and yellow pigments undesirable in the fermentation? To prove that the anti-inflammatory effects are due to amine and amino acid derivatization of orange pigments, the authors need to provide data and analysis involving a proper set of control to compare. Without this key data included, the conclusion is not valid and does not present an issue of fact.

Reviewer 2 Report

Major comments:

Line 26-28 „The derivatives of 2-amino-4 picoline and 4-amino27 4H-1,2,4-triazole exhibited NO inhibitory activities of 74.2% and 63.3%, and cell viabilities of 90.6% 28 and 87.6%, respectively.“ It looks like that the derivatives inhibit rather the growth of the macrophages than NO production. In order to conclude the anti-inflammatory potency, the compounds should reduce NO production together with no cytotoxicity. However, such comparison is in this current version of manuscript almost impossible, I suggest the preparation of the table comparing both activities.

Why anti-inflammatory, non-toxic amino derivatives were excluded from the iNOS experiment? At least D41 as the most active amino derivative?

Line 160 „Measurement of NO Concentration“. There is no mention about NO calibration curve in the methodology. Or is there any other possibility how to quantify NO without the curve?

Figure 2 lacks error bars.

Figures 4 and 5 (and corresponding conclusions) lack the statistical analysis (ANOVA) of the results. Both figures´ legends should contain the concentration of applied compounds.

As the other control, original pigment should be used in order to evaluate the effect of modification on biological activity of original compound.

Western blot could be used as a semi-quantitative method; however, it has to follow the standard procedure (like to determine the linear dynamic range of both control and target protein loading; both target and control protein should be quantified in each sample – D2 sample lacks tubulin, etc.). This validation should be part of the methodology.

Minor comments:

Line 27 rather „inhibited NO production“ than „exibited NO inhibitory activity“  

Lines 97, 108 Shouldn´t be „seed culture“ called rather like a „spore culture“?

Lines 110, 113, 115, 119, 182 rpm should be recalculated as × g (rcf). The first unit is not universal and varies with the radius of the machine.

Line 177 „100 μL“ rather than „One hundred microliters“

Line 180 The composition of “radio-immunoprecipitation assay buffer containing protease and phosphatase inhibitors” should be described

Line 192 „…blocks, that is, orange…“ should be changed to „…blocks – orange…“

Line 276 „anti-microbial, anti-viral, anti-cholesterol, anti-obesity, and anti-melanogenesis“ the hyphen should be written only between two vowels.